# Mathematical modeling of vancomycin release from Poly-L-Lactic Acid-Coated implants

Papon Thamvasupong[1], Kwanchanok Viravaidya-Pasuwat [1,2]*

1 Faculty of Engineering, Department of Chemical Engineering, King Mongkut's University of Technology Thonburi, Bangkok, Thailand, 2 Faculty of Engineering, Biological Engineering Program, King Mongkut's University of Technology Thonburi, Bangkok, Thailand

* kwanchanok.vir@kmutt.ac.th

## Abstract

This study aimed to develop a mathematical model to predict the release profile and antibacterial efficacy of a vancomycin delivery system integrated with poly(L-lactic acid)-coated bone implants specifically designed for bone plates. Using Fickian diffusion principles within an ANSYS-CFX computational fluid dynamic model, we validated the model against our *in vitro* vancomycin release and agar diffusion studies, as well as previously published *in vivo* data, confirming the reliability of the model. The model predictions demonstrated the effectiveness of the system in inhibiting bacterial growth in surrounding tissue with no observed toxicity, with a peak vancomycin concentration of 0.95 mg/ml at 6 hours, followed by a decrease to levels that remained effective for antibacterial activity. Furthermore, a sensitivity analysis revealed that the model is particularly sensitive to the half-life of vancomycin, with a maximum sensitivity index of 0.8, indicating its greater impact on the prediction accuracy than the diffusion coefficient, which has a maximum sensitivity index of 0.5. Therefore, precise input of vancomycin's half-life is critical for accurate predictions. These findings offer substantial support for the efficacy of the local delivery system as a promising therapeutic approach against implant-associated infections.

## Introduction

The systemic administration of antibiotics for tackling implant-associated infections poses significant challenges, prompting the exploration of alternative treatment strategies. Local antibiotic delivery offers a promising strategy for achieving high concentrations of antibiotics, thereby enhancing therapeutic efficacy [1]. Furthermore, it reduces systemic exposure to antibiotics mitigating side effects and the potential for antibiotic resistance [2,3]. Research studies have used polymers as antibiotic reservoirs and delivery vehicles due to their favorable properties, particularly because of their minimal side effects on human cells [4–6].

Our research group previously developed a local delivery system for vancomycin, a widely used antibiotic agent, utilizing a double-layer poly(L-lactic acid) (PLLA) matrix coated onto implant materials [7]. The implementation of a drug-free topcoat allowed the extended release

**Data Availability Statement:** All relevant data are within the manuscript and its Supporting information files.

**Funding:** This work was supported by King Mongkut's University of Technology Thonburi under grant the Petchra Pra Jom Klao Ph.D. Research Scholarship (5/2015); the National Research Council of Thailand under FY2017 Thesis Grants for Doctoral Degree Students (NRCT(A) (DPARB) 8/2017, Appendix 7, No. 4 on the grantee list); and Thailand Science Research and Innovation under grant Fundamental Fund-Basic Research Fund (64-3840-FF). the funders had no role in the study design, data collection and analysis, decision to publish, or preparation of the manuscript.

**Competing interests:** The authors declare no competing interest.

of vancomycin for up to three weeks without an undesirable initial burst release. The release mechanism can be best described using the Korsmeyer—Peppas model, in which Fickian diffusion was shown to be the major factor controlling drug release, with a minor contribution from polymer relaxation. This polymeric delivery system demonstrated substantial antibacterial activity against *S. aureus*, a primary causative agent of orthopedic infection, while showing noncytotoxicity to normal cells. These findings provide further evidence to support the potential of the local delivery system as an effective therapeutic strategy for implant-associated infections.

Before a delivery system can be translated into clinical practice, extensive safety data from animal studies are needed [8]; however, this approach can be costly and time-consuming, leading to delays in the development of innovative delivery systems [9]. Therefore, computational simulations based on real scenarios may play an important role in reducing the number of animal experiments required. Simulations enable researchers to model and predict the behavior of delivery systems in a virtual environment, considering a range of factors that can impact their effectiveness. This valuable approach allows researchers to test various delivery strategies and optimize the design of delivery systems without the need for comprehensive animal testing [10].

The use of computer simulations to predict the performance of new drug delivery systems has been reported in several studies. For example, the impact of three-dimensional intravitreal transport on the distribution of controlled-release drugs within the eye was simulated and studied by Stay et al. [11]. This study emphasized the crucial interplay between convection, diffusion, and geometry in enhancing drug delivery and facilitating more effective ocular pharmaceutical treatments. In another study, Chen et al. utilized computational fluid dynamics (CFD) to predict metformin release from a hydroxypropyl methylcellulose (HPMC) matrix-based extended-release formulation [12]. Their model, which was founded on *in vitro* release kinetics, successfully predicted clinical exposure, an outcome validated through a clinical bioequivalence study. Hussain et al. also employed CFD software to investigate drug design and diffusion profiles during the process of degradation and diffusion in the body [13]. They demonstrated how CFD could provide more illustrative results than traditional laboratory experiments, facilitating process optimization, saving costs and improving troubleshooting, among other advantages. These studies contribute to our deeper understanding of drug release and transport mechanisms in biological systems.

In this study, we aimed to construct a mathematical model capable of accurately predicting the concentration profile of vancomycin released from double-layered PLLA-coated bone implants used for bone plates. We employed the data from our previous research to construct this model based on the principle of Fickian diffusion using the computational fluid dynamics (CFD) model in ANSYS-CFX to simulate the vancomycin release process from this unique delivery system. Our model underwent rigorous validation using a two-step process. Initially, we compared our simulation results with the experimental data derived from an *in vitro* vancomycin release study and an agar diffusion assay reported in our previous work [7]. The model was further validated by adapting it to incorporate physiological data from an animal study by Loc-Carrillo et al., who explored the distribution of vancomycin when it was locally injected into the cancellous bone of rats [14]. This exercise facilitated an evaluation of our model's ability to predict the vancomycin concentration profile under real physiological conditions. Upon validation, we extended our model to incorporate human physiological data, simulating the local delivery of vancomycin from a PLLA-coated bone plate. The anticipated outcomes of this novel simulation approach should offer a comprehensive understanding of vancomycin behavior within the human body, thereby contributing to the prediction of the effectiveness of this local delivery system in preventing implant-associated infections. Finally,

we performed a sensitivity analysis to examine the reliability of our simulation results. We allowed for variations in key simulation parameters such as the diffusion coefficient of vancomycin and its half-life, as per the literature values. This step was critical for ensuring the accuracy of our results, as it revealed the impact of these parameters on the concentration profile of vancomycin in the body if the actual parameters deviated from those of our model.

## Numerical method for transient diffusion

Building upon our previous research, the release of vancomycin from a double-layer PLLA-coated substrate was predominantly governed by Fickian diffusion, a mechanism whereby drug dispersion was driven by a concentration gradient [7]. An exact solution to these one-dimensional diffusion problems can be obtained by Fick's first law of diffusion and Fick's second law of diffusion (Eqs 1 and 2).

$$J = -D\frac{dC}{dx} \tag{1}$$

$$\frac{\partial C}{\partial t} = D\frac{d^2C}{dx^2} \tag{2}$$

where $J$ is the mass flux (mg·cm$^2$·h$^{-1}$)
$D$ is the diffusion coefficient (cm$^2$·h$^{-1}$)
$C$ is the concentration of vancomycin (mg·cm$^{-3}$)
$x$ is the distance in x-direction (cm)
$t$ is time (h)

However, when tackling a more complex problem, deriving an exact solution analytically can prove challenging. As a result, numerical solutions offer a practical alternative. In this study, we employed a numerical solution method, illustrated in S1 to calculate the system's behavior at each time step. S1 depicts the diffusion of vancomycin from the delivery system (positioned on the left side of the diagram) into the body fluid interface, represented by $i$ equal to 0. The process of diffusion extends across every element, from the leftmost to the rightmost side. Eqs 3 and 4 describe the mass balance of vancomycin in this context.

$$\text{At } i = 0; \quad \frac{\Delta m_0}{\Delta t} = AJ_{\text{in}} - AJ_{\text{diffusion}} \tag{3}$$

$$\text{At } i = 1, 2, 3, \ldots, \text{n}; \quad \frac{\Delta m_i}{\Delta t} = AJ_{\text{diffusion,in}} - AJ_{\text{diffusion,out}} \tag{4}$$

A simple half-life equation was incorporated into the model to estimate the clearance rate at which the body removes vancomycin from each individual element. The equation for the half-life ($t_{1/2}$) is shown in Eq 5.

$$m_t = m_0 \left(\frac{1}{2}\right)^{t/t_{1/2}} \tag{5}$$

Ultimately, the concentration of vancomycin within each element can be estimated using

the equations below (Eqs 6–8). For a comprehensive derivation, please refer to S2.

$$C_0|_{t+\Delta t} = C_0|_t + \frac{\Delta t}{\Delta x^2}[\Delta x \cdot J_{in} + D(C_1|_t - C_0|_t)] + \left[\frac{\Delta t \cdot \ln(0.5)}{t_{1/2}} \cdot C_0|_t \cdot (0.5)^{\Delta t/t_{1/2}}\right] \quad (6)$$

$$C_i|_{t+\Delta t} = C_i|_t + \frac{D\Delta t}{\Delta x^2}\left[C_{i-1}|_t - 2C_0|_t + C_{i+1}|_t\right] + \left[\frac{\Delta t \cdot \ln(0.5)}{t_{1/2}} \cdot C_i|_t \cdot (0.5)^{\Delta t/t_{1/2}}\right] \quad (7)$$

$$C_n|_{t+\Delta t} = C_n|_t + \frac{D\Delta t}{\Delta x^2}\left[C_{n-1}|_t - C_{nt}\right] + \left[\frac{\Delta t \cdot \ln(0.5)}{t_{1/2}} \cdot C_n|_t \cdot (0.5)^{\Delta t/t_{1/2}}\right] \quad (8)$$

Since there are no available data on the diffusion coefficient of vancomycin through bone, we extrapolated the effective diffusion coefficient ($D_{eff}$) using Eq 9 [15]. This calculation relies on the diffusion behavior of vancomycin in water and the inherent properties of the bone structure, including constrictivity, porosity, and tortuosity. The reported values for constrictivity in cortical bone and cancellous bone were 0.84 and 0.96, respectively [15]. The porosities of the cortical bone and cancellous bone were 0.05 and 0.95, respectively (we chose the higher end of the range for cancellous bone, 0.75–0.95, for elderly patients who had decreased bone mass) [16–18]. The tortuosity values for cancellous bone were reported to be in the range of 1.02–1.70 [19–21]. However, the tortuosity of cortical bone was not available in the literature. Therefore, we assumed that the tortuosity of cortical bone was the same as the average tortuosity of cancellous bone, which was 1.10.

$$D_{eff} = D_0 \frac{\delta\varphi}{\tau} \quad (9)$$

where $D_0$ is the free water diffusion coefficient of vancomycin at 37°C of
$2.83 \times 10^{-6}$ cm$^2$/s [22]
$\delta$ is the constrictivity factor
$\varphi$ is the porosity of the bone
$\tau$ is the tortuosity factor

To ensure numerical stability during the calculation, a sufficiently small time step size was needed. The time step size selection was selected based on the Courant—Friedrichs—Lewy rule, as described in previous studies [23] and shown in the equation below.

$$\Delta t \leq \frac{\Delta x^2}{6D} \quad (10)$$

where $\Delta t$ is the time step size (s)
$\Delta x$ is the calculation element size (mm)
$D$ is either the diffusivity of vancomycin in tissue or the effective diffusion coefficient in bone (mm$^2$ s$^{-1}$)

## Materials and methods

### Simulation setup and process

The simulation for this study was conducted using the finite element software ANSYS-CFX 2021R. The simulation process was organized into five distinct modules. First, the geometry of the model was constructed using the Design Modeler module. Subsequently, the computational mesh was generated in the Mesh module. In the CFX-Pre module, all necessary physical

models, boundary conditions, and initial conditions were configured. The concentration of vancomycin was defined as an additional volumetric variable, which is essential for the implementation of Eqs 6–8 in the simulation. To ensure continuous release into the domain, a sink condition was applied. The boundary condition of a no slip wall was set at the release surface, whereas the other faces of the domain were treated as symmetric. The CFX-Solver module was then employed for the calculations. Upon completion of the simulation, the CFX-Post module was utilized to visualize and analyze the results.

## Validation of the simulation model

Our simulation model underwent a two-part validation process. The first part involved comparing simulation outputs with the experimental data from our prior *in vitro* vancomycin release study and agar diffusion assay [7]. The second part of the model validation involved conducting a comprehensive simulation of vancomycin release from a bone plate, which was subsequently compared with the *in vivo* data reported by Loc-Carrillo, et al. [14].

**In vitro vancomycin release study.** Double-layered PLLA-coated substrates containing vancomycin were incubated in simulated body fluid at 37ºC. The vancomycin concentrations released from the coated substrate were analyzed at predetermined intervals over a 20-day period to calculate the cumulative release percentage. To simulate this experiment, several assumptions were made: diffusion was considered one-dimensional, a sink condition was assumed for the receiving medium, and vancomycin degradation was not considered. The diffusion coefficient of vancomycin in water, $2.83 \times 10^{-6}$ cm$^2$/s [22], was used in this simulation.

**Agar diffusion experiment.** In this experiment, a $3 \times 3$ cm$^2$ substrate coated with a PLLA layer containing vancomycin was placed on an *S. aureus* culture agar plate. After a 24-hour incubation, the extent of the bacterial inhibition zone was measured. To create this experimental set up in a simulated environment, we constructed a $3 \times 3$ cm$^2$ square in the CFD (Fig 1a) and imposed a mesh grid outside this square, initially with a mesh size of 0.1, to simulate the fluid domain (Fig 1b). The diffusivity of vancomycin in the agar medium, set to 0.72 mm$^2$/h ($2.08 \times 10^{-6}$ cm$^2$/s), was estimated based on the work of Cavenaghi, et al. [24]. Given that the agar medium did not facilitate vancomycin clearance, the half-life of vancomycin was not included in this particular simulation. We assumed symmetrical and no-slip boundary conditions. To represent the inhibition zone in the simulation, we identified areas within the fluid domain where vancomycin concentrations exceeded the minimum inhibitory concentration (MIC) of *S. aureus*, 2 µg/ml [25]. Finally, to ascertain the accuracy of the simulation, we also conducted a mesh dependency test (S3).

## Validation with previously published in vivo data

The vancomycin concentration data (in µg/ml) in rat tibias, provided in the S1 File of the study by Loc-Carrillo, et al. [14], served as a reference point for this comparison. Despite the variation in experimental setups, where Loc-Carrillo et al. used local injection of vancomycin into the bone and, in our model, vancomycin was locally delivered from a bone plate coated with a vancomycin-PLLA layer, it was still possible to make meaningful comparisons. The concentration of vancomycin in the simulation was set to match the injection dose used in the animal study. Because the half-life of vancomycin in rats has not been reported in the literature, the half-life of vancomycin is assumed to be 4 hours, which is the same as the half-life of vancomycin in humans [26]. Our attention specifically focused on the concentration profile of vancomycin within the cortical bone, a parameter shared between the two studies. We compared the distribution of vancomycin within the cortical bone over time between our simulation and

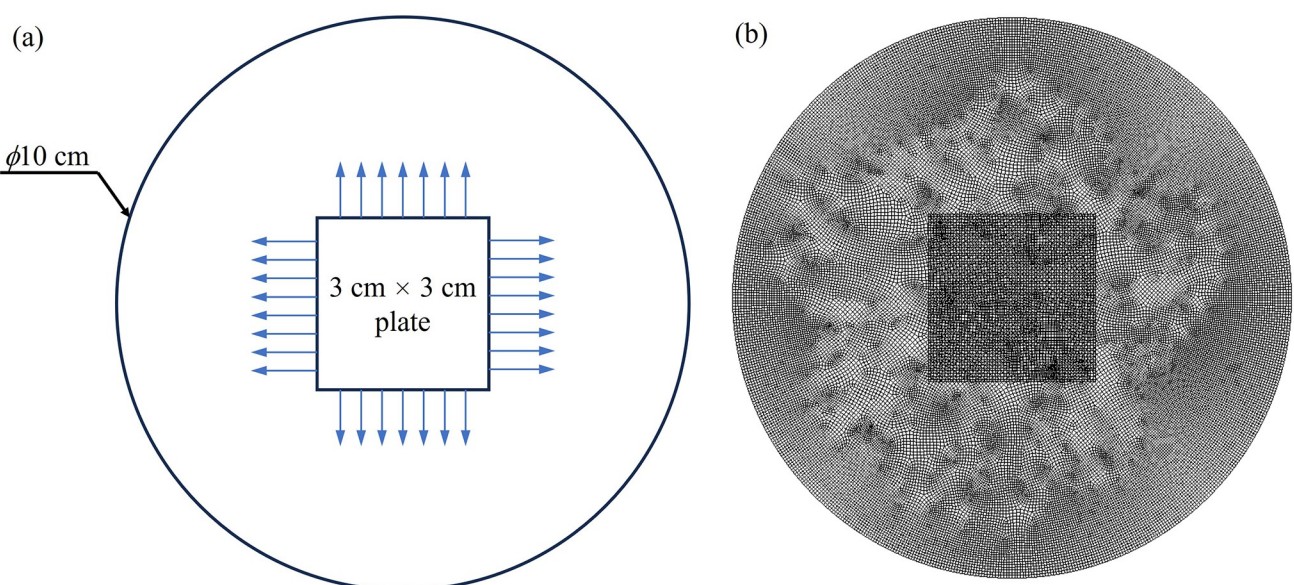

**Fig 1. The fluid domain for agar diffusion simulation: (a) schematic representation and (b) mesh used in the simulation.**

that in the animal study. The configuration of the fluid domain used for this simulation is detailed in Fig 2.

## Simulation of the delivery system

We applied our model to a case study to simulate the dispersion of vancomycin from a bone plate coated with a PLLA layer containing antibiotics. As the bone plate is designed to allow the antibiotic to diffuse in two directions, as shown by the blue arrows in Fig 3, vancomycin can permeate both the surrounding bone and body tissue, the two main sites susceptible to bacterial infection. To simplify our computational model, we omitted the edge effect in the

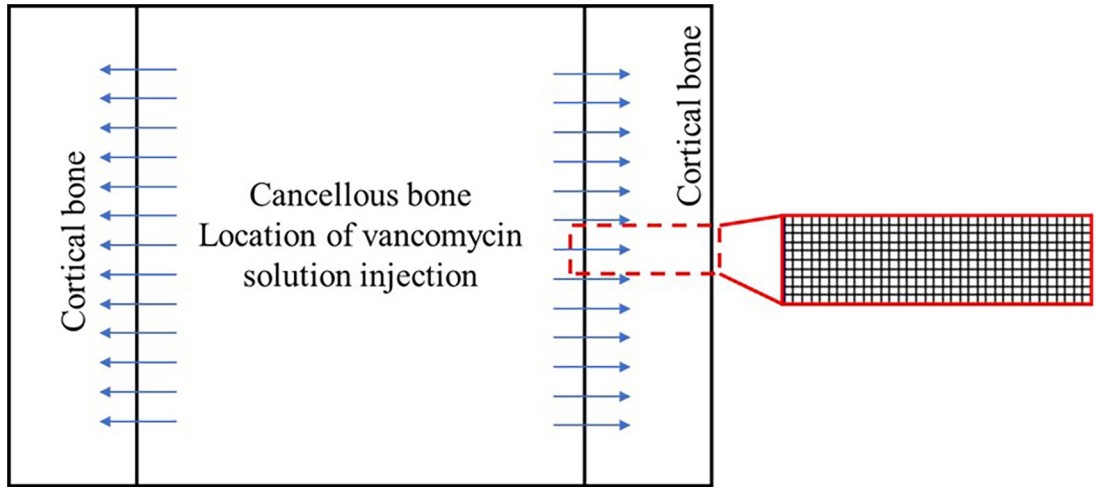

**Fig 2. Conceptual illustration and mesh of the fluid domain for validation compared with those of the animal study.**

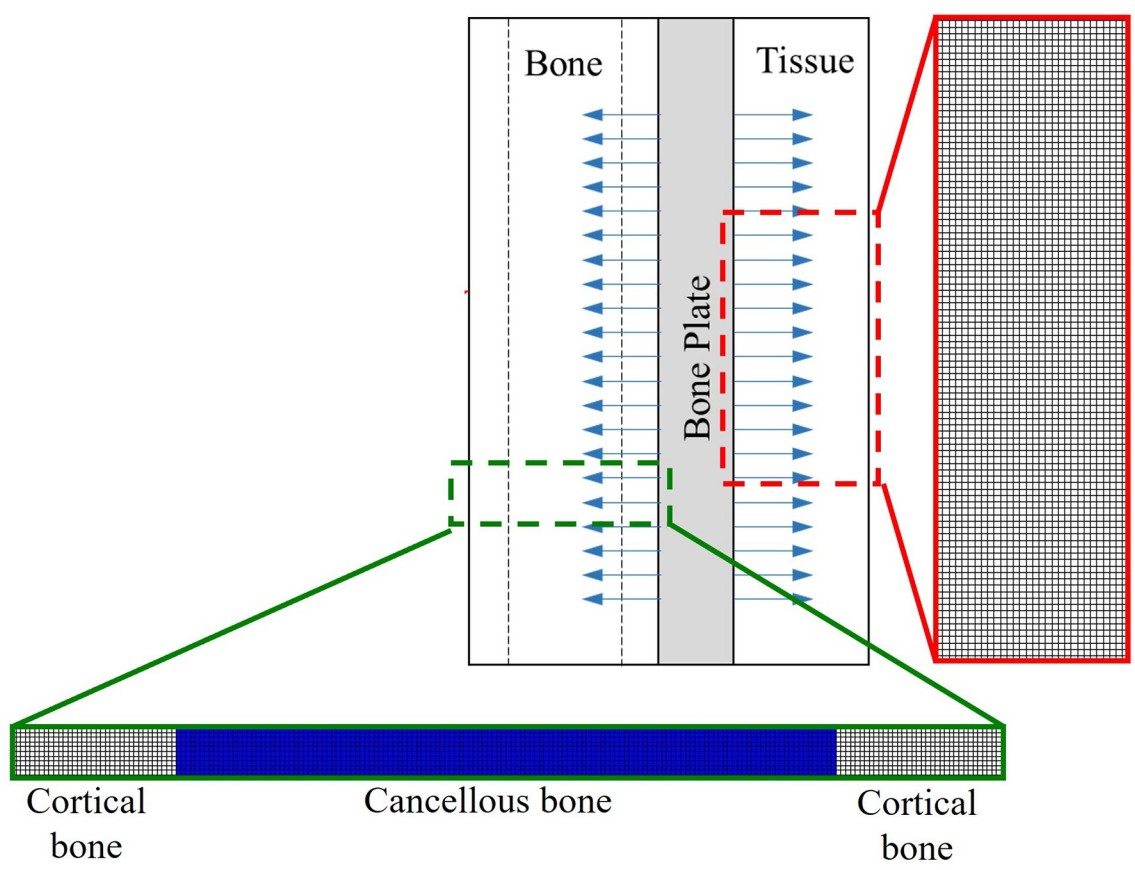

**Fig 3. Conceptual geometry and mesh for the bone plate simulation.**

simulation, thereby enabling vancomycin to disperse uniformly across the full extent of the tissue and bone.

For the tissue side, we modeled the domain as a rectangular volume measuring 3 mm in diffusion length to minimize the simulation size and 10 mm in height to eliminate edge effects at the midpoint, as shown in Fig 3. A diffusion distance of 3 mm was deemed sufficient, as bacterial infections typically occur near the bone plate during the early postoperative period [27]. On the bone side, the fluid domain was divided into three subdomains, consisting of two areas of cortical bone and a region of cancellous bone. Each subdomain was established with a square cross-sectional area of $1 \times 1$ mm$^2$, while the lengths of the subdomains were derived from the anatomical dimensions of human cortical and cancellous bones, measuring 3.5 mm and 14 mm, respectively [28]. The diffusion coefficient of vancomycin through tissue was approximated to be similar to its diffusion through agar, $2.08 \times 10^{-6}$ cm$^2$/s, as proposed by Cavenaghi, et al. [24]. Furthermore, the effective diffusion coefficients ($D_{\text{eff}}$) of vancomycin, computed from Eq 9 were $2.26 \times 10^{-7}$ cm$^2$/s for cortical bone and $2.24 \times 10^{-6}$ cm$^2$/s for cancellous bone.

The boundary conditions for the models were established such that the vancomycin influx was derived from experimental data from our *in vitro* release study [7]. All the other boundaries were set as having no gradient at the interfaces. For the body clearance of vancomycin, a domain negative source expression was used. In addition, the transition between cortical and

cancellous bones was modeled without any diffusive resistance, maintaining equal concentrations of vancomycin across these interfaces.

## Parameter sensitivity analysis

A sensitivity analysis was carried out by adjusting key parameters from their base values, including the diffusion coefficient of vancomycin in tissue ($2.0 \times 10^{-6}$ cm²/s), the half-life of vancomycin in the human body (4 h), and bone porosity (0.05 for cortical bone and 0.95 for cancellous bone). This analysis was based on data extensively collected from the literature [15–18,29–36]. After ensuring a normal distribution of the data, we estimated the average and standard deviation for all the parameters, which were subsequently log-transformed. We generated random scaling factors using Microsoft Excel based on the standard deviation of the published values. The distribution of these 50 random points is detailed in S4. Of these, eleven scaling factors were selected to modify the baseline values. These values were chosen to represent percentile rankings from 0 to 100 in increments of 10. Subsequently, we employed a statistical linear regression method to evaluate the changes in the vancomycin concentration. If the $R^2$-adjusted value closely matches the $R^2$ value, the number of selected points could be reduced to eight. Finally, a sensitivity index was determined using Eq 11

$$Sensitivity\ Index\ = \frac{Output\ change\ percentage}{Input\ change\ percentage} \tag{11}$$

## Results and discussion

### Model validation with *in vitro* vancomycin release and agar diffusion studies

The first step in validating our simulation model was to compare its predictions with the experimental data from our previous *in vitro* studies on vancomycin release and antibacterial activity using an agar diffusion assay. As shown in Fig 4a, the cumulative release profile of vancomycin from the double-layered PLLA-coated substrate predicted by our simulation closely aligns with the experimental data, demonstrating the model's accuracy. In the agar diffusion

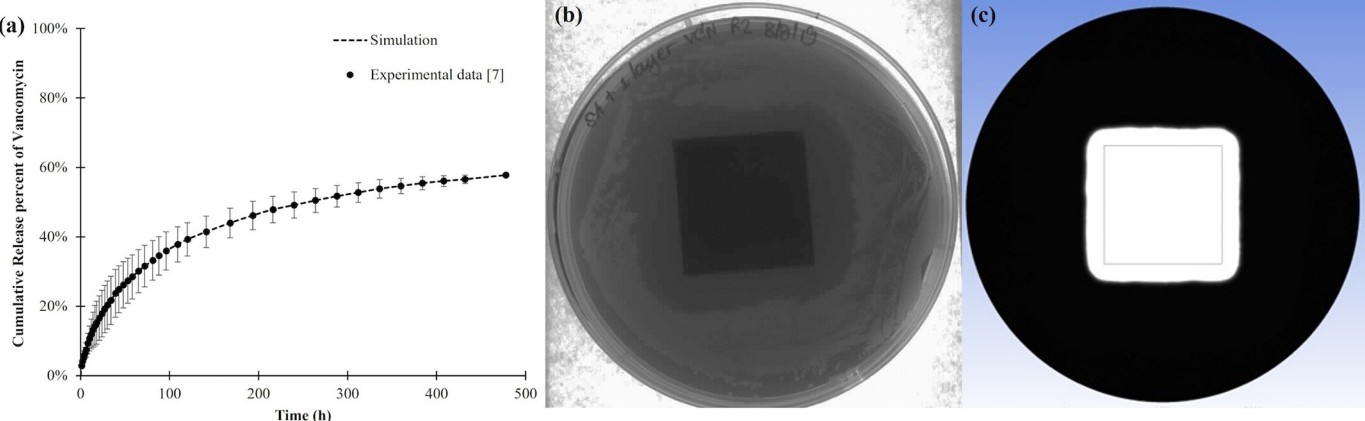

**Fig 4. Validation of the model predictions against our *in vitro* experimental results: (a) *in vitro* vancomycin release study [7] and agar diffusion assay: (b) experiment [7] and (c) the predicted inhibition zone.**

study (Fig 4b), the experimentally determined inhibition area of the vancomycin-PLLA coating on the substrate was 12.97 ± 1.30 cm$^2$ [7]. In the simulation, as illustrated in Fig 4c, the white zone represents the region where the vancomycin concentration exceeded the minimum inhibitory concentration (MIC) of $2 \times 10^{-6}$ mg/mL, indicating the capacity of vancomycin-PLLA coating to inhibit bacterial growth. The computationally derived area of this inhibition zone was approximately 15.43 cm$^2$, which was slightly larger than the inhibition area observed in the agar diffusion assay. A t-test statistical analysis was used to compare the bacterial inhibition areas yielded by the simulation and the experiment, the results of which yielded a p-value of 0.082 (significance level: 0.05). This outcome suggested that the areas of bacterial inhibition observed in the simulation and the experiment were comparable. Therefore, these findings support the reliability and precision of the model used in the simulation.

## Cross-validation with previously published *in vivo* data

In this phase of the validation process, we compared our simulation results against those of an existing *in vivo* study conducted by Loc-Carrillo et al. [14]. Despite the differences in the experimental setup, where vancomycin solution was injected directly into the rat's cancellous bone in the *in vivo* study versus a continuous supply to the cortical bone in our simulation, we managed to bridge the gap by setting comparable initial drug concentrations. As shown in Fig 5, which shows the average concentration of vancomycin in the tibial cortex over time, there was an overlap between the concentration profiles from the *in vivo* study and our simulation data. Both profiles display an initial rapid increase in vancomycin concentration within the first six hours, followed by an exponential decrease thereafter. These similarities can be attributed to the equal initial vancomycin doses used in both our simulation and the *in vivo* study. However, the exact concentration values varied. The simulation indicated a higher vancomycin concentration than what was observed in the animal study. This discrepancy might be explained by potential inaccuracies in the half-life of vancomycin assumed for our simulation. The typical half-life of vancomycin in rats is reported to be between 0–6 hours [37]. The assumed half-life of vancomycin used in the simulation was 4 hours, which might be overestimated, leading to slower clearance in the simulation model than what occurred in reality. These findings emphasize the importance of accurate assumptions regarding drug pharmacokinetics, such as the half-life, when developing simulation models. This finding also highlights the potential need for further refinement of the simulation model to account for these complexities. Despite the differences in absolute concentrations, the overall trend similarity between the *in vivo* study and the current study suggests that the simulation is still promising and that our model can be a useful tool for predicting vancomycin diffusion dynamics in bone.

## Model application: Predicting antibiotic efficacy of vancomycin-PLLA coated bone plates

The validated model was used to predict the antimicrobial efficacy of bone plates coated with vancomycin-PLLA. Since the bone plate is placed at the interface between muscle tissue on one side and bone on the other, it is crucial to understand the distribution of vancomycin at both interfaces to assess the efficacy of antibiotic treatment in the event of bacterial infection associated with the implant. Our model provides insight into antibiotic distribution, elucidating whether the resulting concentration of vancomycin is adequate for inhibiting bacterial growth, while being nontoxic to human cells.

**Time-dependent distribution of vancomycin from the bone plate to muscle tissue.** The concentration profile of vancomycin across the muscle tissue domain exhibited dynamic

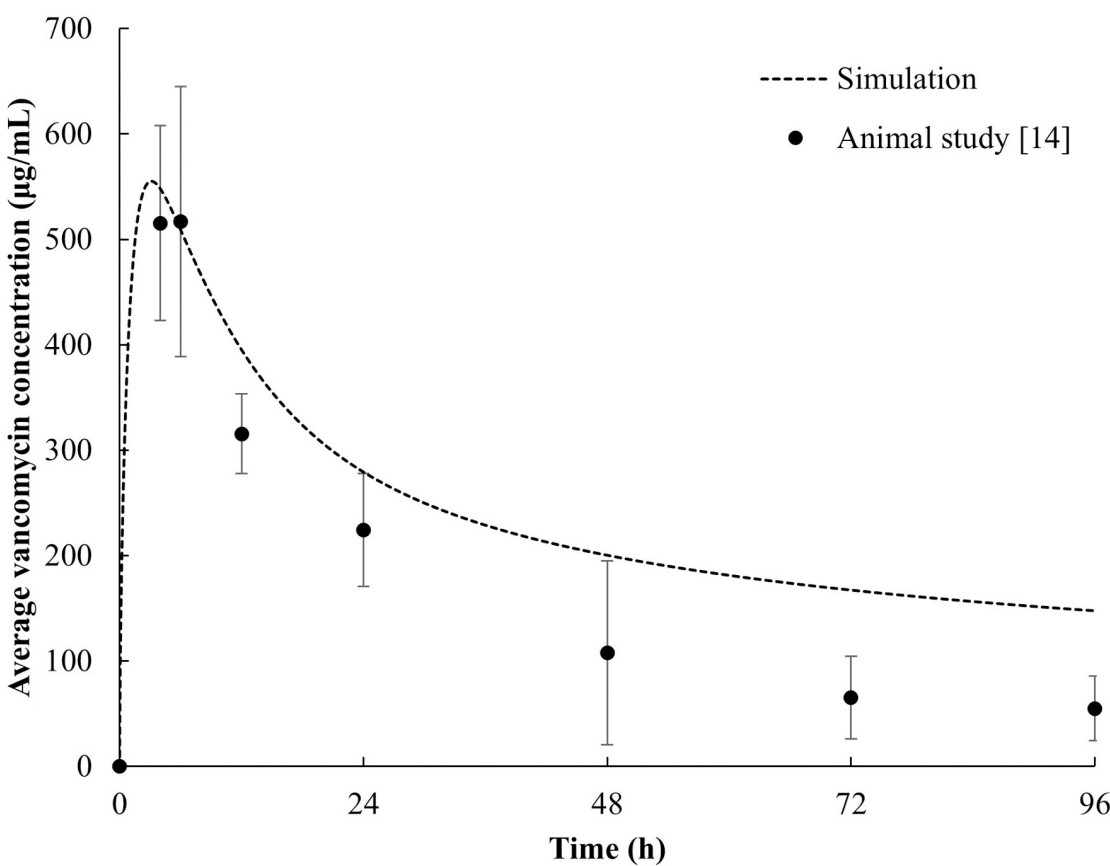

**Fig 5. Average concentration of vancomycin in the cortical bone of rat tibias at different time points as predicted by our simulation model, compared with the *in vivo* experimental data from previously published literature [14].**

behavior post-implantation of the vancomycin-PLLA coated bone plate. The highest average concentration of vancomycin across the entire domain reached approximately 950 µg/mL at 6 hours (Fig 6). However, a significant decrease in the vancomycin concentration was observed approximately 12 hours after the simulation, particularly near the bone plate. This decrease can be attributed to the pharmacokinetic properties of vancomycin, where the drug clearance rate, governed by its half-life, surpassed the release rate from the bone plate. As a further observation, 24 hours into the simulation, a consistent decrease in vancomycin concentration was recorded, supporting the earlier finding of accelerated clearance after the 12-hour mark. The vancomycin concentration at all the simulated time points exceeded the minimum inhibitory concentration (MIC) of 2 µg/mL, suggesting the potential for sustained inhibition of bacterial growth throughout the considered timeframe.

Vancomycin has been shown to inhibit cell viability and functionality across various cell types, including osteoblasts, endothelial cells, fibroblasts and skeletal muscle cells [38]. The reduction in cell proliferation was found to be both dose- and time-dependent [39]. Significant inhibition was observed at vancomycin concentrations of 2 mg/mL in all the examined cell types [38]. Specifically, fibroblasts and skeletal muscle cells exhibited the highest sensitivity to vancomycin, particularly at concentrations exceeding 1 mg/mL after 3 days of exposure [38]. In our simulation, the maximum predicted concentration of vancomycin released from the bone plate into the adjacent tissue remained well below the toxicity threshold of 1 mg/mL [40],

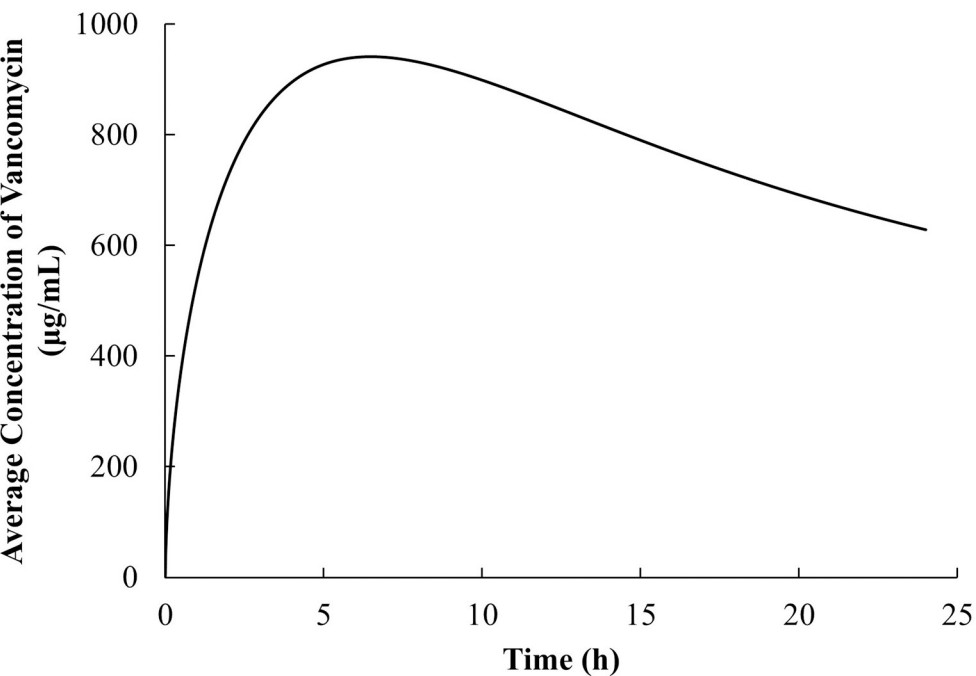

**Fig 6. Average concentration of vancomycin over the first 24 hours of the CFX simulation in the case of a bone plate.**

peaking at 6 hours post-implantation (Fig 6). In addition, the predicted vancomycin concentration steadily decreased over time, further minimizing potential toxic effects on surrounding tissue. These findings suggest that the use of vancomycin-coated bone plates, as modeled in our study, is unlikely to induce significant toxicity *in vivo*. Nonetheless, it is important to recognize that the referenced toxicity thresholds were derived from *in vitro* experiments, which may not fully replicate *in vivo* conditions, including tissue perfusion, the presence of serum proteins, and enzymatic activities that could influence both the concentration and toxicity profile of vancomycin *in vivo*.

As discussed in the previous section, the half-life of a drug significantly influences its systemic distribution and clearance rate. The typical half-life of vancomycin in healthy adults ranges from 4 to 6 hours [26]. Therefore, in this study, we conducted simulations using half-lives of 0, 4, and 6 hours to investigate the impact of these half-lives on vancomycin distribution in muscle tissue adjacent to the bone plate (Fig 7). Without accounting for vancomycin's half-life (i.e., when the half-life was set to zero), we observed a continual increase in the drug's concentration, as there was no mechanism in place to simulate drug clearance from the system. Upon integrating the half-life into the model, the simulation demonstrated a peak in the vancomycin concentration, followed by a decrease. This pattern aligns with a scenario in which the body's clearance of the drug eventually surpasses the rate of drug release from the bone plate. A longer half-life resulted in a slower reduction in vancomycin concentration in the tissue, as observed when comparing the 4-hour and 6-hour half-life simulations. These observations emphasize the critical role of vancomycin's half-life in predicting the drug distribution and pharmacokinetic behavior of antibiotic-coated bone plates.

Previous studies have demonstrated that vancomycin follows a bi-phasic elimination pattern after intravenous administration, characterized by a rapid initial half-life and a terminal

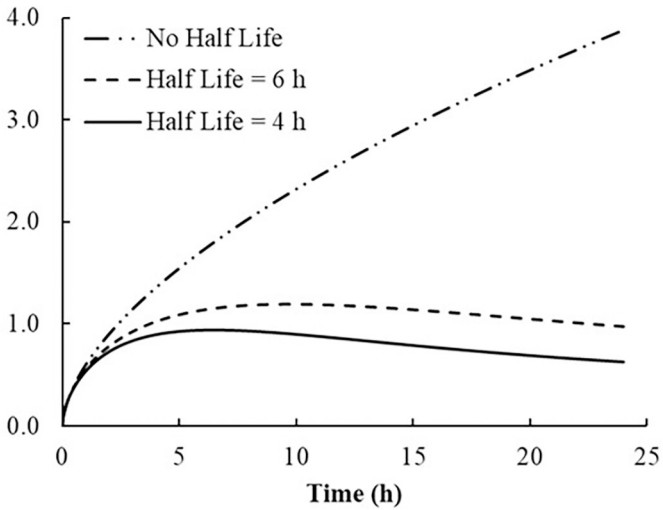

**Fig 7. Average concentration of vancomycin released from the bone plate into the tissue with different half-life scenarios, (- · ·) no half-life, (- - -) half-life of 4 hours, and (- - -) half-life of 6 hours.**

half-life ranging from 4.7 to 11.2 hours in healthy adults with normal renal function [36]. However, another study suggests that the pharmacokinetics of vancomycin may be better described by a triexponential model. This model proposes an initial half-life of approximately 7 minutes, followed by a second half-life of 0.5 to 1 hour, and a terminal elimination half-life ranging from 3 to 9 hours in individuals with normal renal function [41]. In addition, vancomycin clearance is influenced by age, with older adults exhibiting a longer estimated terminal phase half-life than younger adults (12.1 hours vs. 7.2 hours), which slows the drug elimination, suggesting the necessity of dose adjustments for elderly patients [42]. These reports highlight the complexity of vancomycin pharmacokinetics, particularly in terms of elimination. Therefore, incorporating patient-specific pharmacokinetic parameters is crucial for accurately predicting the concentration profile of vancomycin in our model.

**Time-dependent distribution of vancomycin from the bone plate to bone tissue.** The composite structure of bone tissue, comprising the outer cortical bone and the inner cancellous bone, presents unique obstacles to the diffusion of vancomycin, owing to the differences in porosity and diffusion coefficient. As shown in Fig 8a, the dense structure of the cortical bone significantly restricted the diffusion of vancomycin to the cancellous bone region. Vancomycin can only permeate through the microscopic pores within the bone, which leads to a slower drug distribution rate and subsequently an exponential decrease in the vancomycin concentration near the bone plate. Due to this diminished concentration gradient, the cancellous bone receives only a limited vancomycin supply.

Fig 8b shows a visual interpretation of the regions with effective antibacterial activity. Here, the vancomycin concentration contours are calibrated to illustrate a range from 2 μg/mL (the minimum inhibitory concentration, highlighted in green) to 1,000 μg/mL (the toxicity threshold, highlighted in red). Areas where vancomycin concentrations fall below 2 μg/mL are shown in blue. A high concentration of vancomycin was predicted in the cortical bone area adjacent to the bone plate, with some regions even exceeding the toxicity threshold of 1,000 μg/mL (Fig 8b). Although the concentration surpasses the toxicity level, it is rapidly

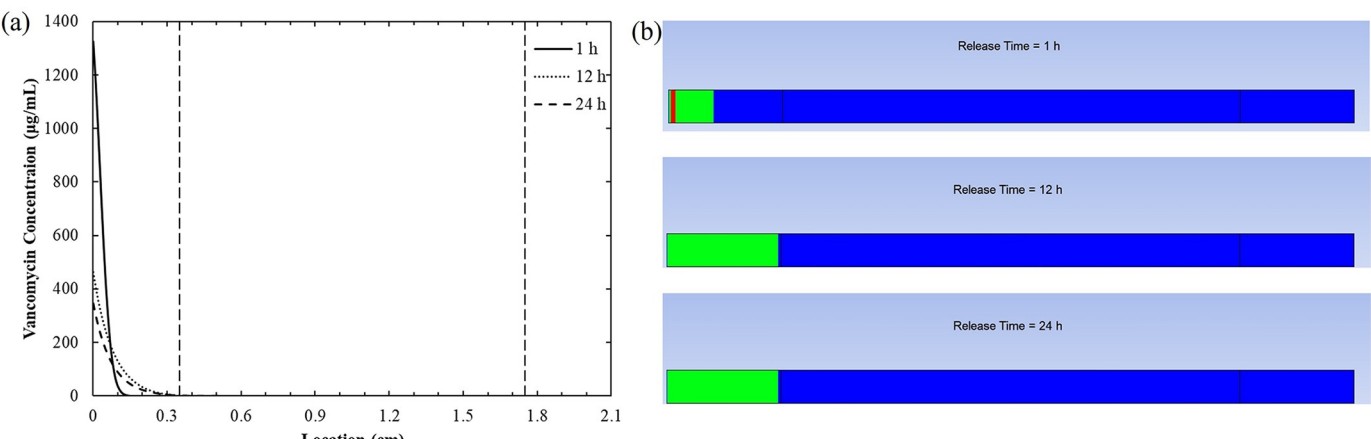

**Fig 8. (a) Vancomycin concentration profiles at 1, 12, and 24 hours (b) Contour map of vancomycin concentrations: Blue represents regions of insufficient antibiotic activity, green denotes zones of effective antibacterial activity, and red indicates potential areas of vancomycin toxicity.**

depleted in the subsequent hours. This suggests that the potential for damage to surrounding tissues from high vancomycin exposure, specifically nephrotoxicity, is minimized as it typically requires exposure over several days [43]. However, the primary concern lies near the bone plate, where bacterial infections typically form biofilms [44]. Hence, maintaining high vancomycin concentrations in this vicinity is crucial for preventing biofilm formation. Suboptimal antibiotic activity in the cancellous bone region is less of a concern as infections predominantly occur in tissues in close proximity to the bone plate. As a result, the findings indicate that the bone plate delivery system, by facilitating the diffusion of vancomycin within the bone, could provide a safe and effective approach to maintain antibacterial activity.

Bone tissue is composed of approximately 60% inorganic material (primarily hydroxyapatite), 10% water and 30% organic components, mainly proteins [45]. Several antimicrobial agents bind to hydroxyapatite, resulting in reduced drug penetration into bone tissue [46]. Vancomycin is known to adsorb to the micropores of hydroxyapatite and is subsequently released into the surrounding fluid over a period of 5 days [47]. This binding interaction could alter the release profile of vancomycin from the bone plate as predicted by our simulation. Adsorption to hydroxyapatite may lead to the accumulation of vancomycin at the interface between the bone plate and the bone, which could result in a concentration that may be detrimental to nearby bone cells. However, studies have demonstrated that vancomycin has a good penetration profile into bone tissue [48], suggesting that its binding to hydroxyapatite may not significantly impact the concentration profile as predicted. To enhance the accuracy of the model, incorporating the binding dynamics of vancomycin to hydroxyapatite would be beneficial.

## Sensitivity analysis

A sensitivity analysis was conducted to assess the impact of varying key parameters on the concentration of vancomycin, a critical factor in the model outcomes (Fig 9). The parameters examined in this sensitivity analysis included the diffusion coefficients of vancomycin in human tissue and cortical bone, as well as the half-life of vancomycin. As data for the diffusion coefficients of vancomycin in human tissue and cortical bone were unavailable, these two parameters were estimated based on the diffusivity of vancomycin in agar and anatomical data regarding cortical bone reported in the literature, respectively. According to Fig 9, the

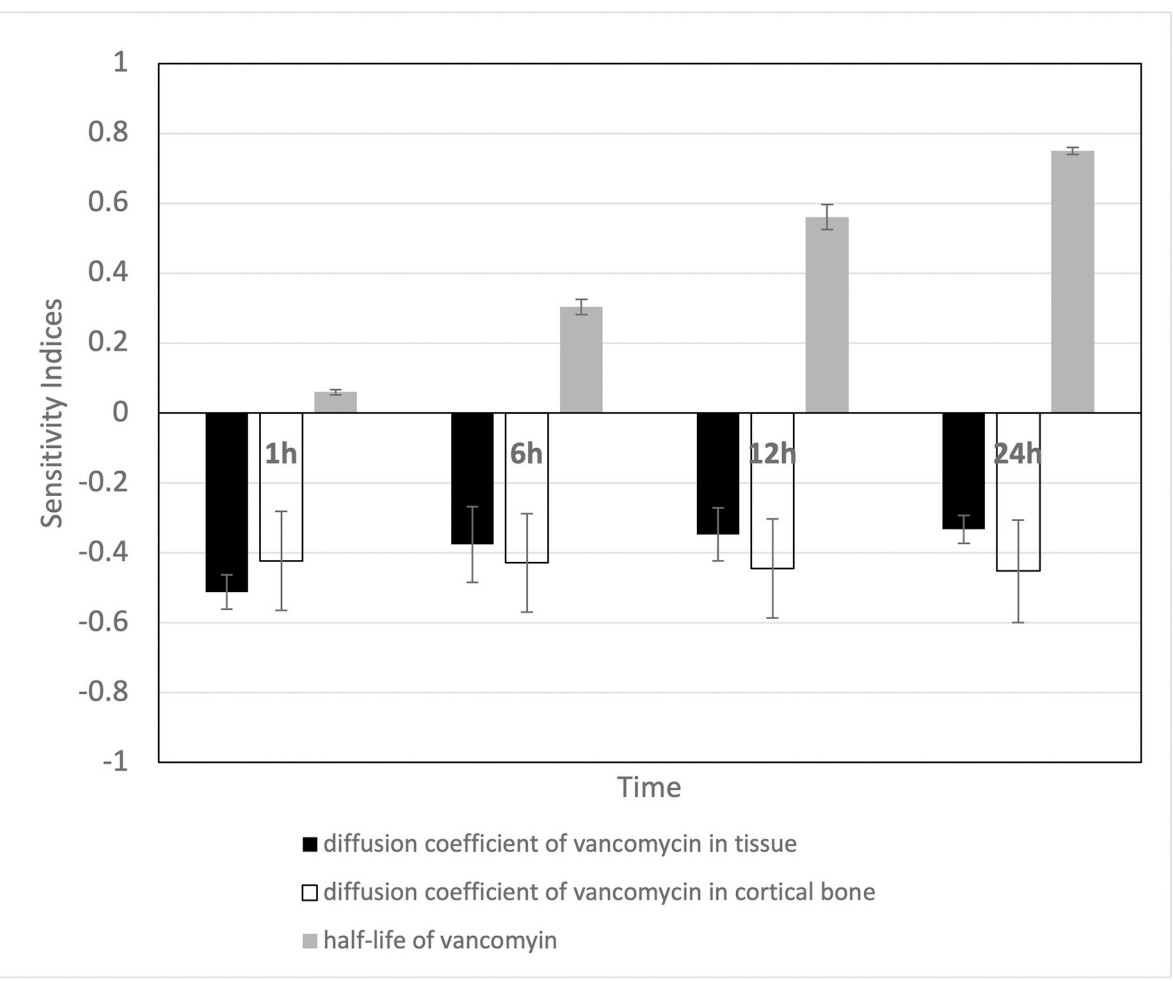

**Fig 9. Sensitivity indices of the diffusion coefficients of vancomycin in tissue and cortical bone and the half-life of vancomycin.**

diffusion coefficients of vancomycin in both tissue and cortical bone exhibit a negative relationship. An increase in the diffusion coefficient of vancomycin results in a lower concentration of vancomycin in the surrounding areas, as a higher diffusion coefficient allows vancomycin to disperse further. A 20% change in the diffusion coefficient of vancomycin, as observed in the literature, corresponds to a 6–10% change in the predicted vancomycin concentration, which is not considered highly significant.

The half-life of vancomycin was varied between 4–6 hours, with a 4-hour half-life used in the model's previous calculations. A longer vancomycin's half-life leads to a slower elimination rate of vancomycin, resulting in a higher concentration and a positive sensitivity index. According to Fig 9, the half-life of vancomycin is also shown to be time sensitive. In the first hour, changes in vancomycin's half-life have minimal impact on the concentration predictions, as only a small amount of vancomycin has been eliminated. However, the effect becomes more pronounced over time, particularly at 24 hours, where a 50% increase in the half-life (from 4 to 6 hours) results in an approximately 37% change from the nominal value. This substantial variation could pose a risk of toxicity in the surrounding tissue, particularly in patients with renal dysfunction, who would have a prolonged vancomycin's half-life [26]. As a result, close therapeutic drug monitoring is necessary for these patients.

### Limitations

While the simulation confirmed the effectiveness of the vancomycin delivery system in maintaining antibacterial efficacy, it is important to acknowledge certain limitations in our model's ability to predict the release of vancomycin from coated implants. Real-world conditions can exhibit significant variability, including differences in patient physiology, medical history, age and other factors that may influence drug release and clearance. Our model is particularly sensitive to the half-life of vancomycin. A small variation in this parameter could lead to a significant change in the predicted outcome, emphasizing the need for accurate input data. Furthermore, our model incorporates certain assumptions and simplifications, which may not fully capture the complexity of *in vivo* processes. For example, we assumed that diffusion into tissue and bone occurred without interface resistance. Moreover, the interaction between vancomycin and bone's inorganic components, such as hydroxyapatite, was not considered, which could lead to deviations in the actual vancomycin concentration from our predictions. This variation might affect both the rate of vancomycin penetration into bone and its therapeutic efficacy. While our model provides valuable insights, it should be interpreted with caution in real-world patient cases. Further studies with animal or clinical data could enhance the reliability and clinical relevance of our mathematical model.

## Conclusions

Our simulation using ANSYS CFX to predict the vancomycin release profile from a PLLA coating has been successfully validated with an agar diffusion assay and an animal study. The results revealed that this system has the potential to inhibit bacterial growth in the surrounding tissue without inducing toxicity. While diffusion through bone presented limitations for inhibiting bacterial growth within bones, it was observed that infections primarily occurred on the surface, where the system demonstrated excellent antibacterial activity. A sensitivity analysis was carried out to identify influential variables affecting the model's predictions. Changes in the half-life of vancomycin had a more substantial impact on the model's predictions than did variations in the diffusion coefficient of vancomycin.

These findings collectively contribute to our understanding of the efficacy of the vancomycin delivery system and its potential applications in preventing implant-associated infections. Further studies can focus on refining the model and exploring additional factors to optimize the antibacterial performance of the system, ensuring effective treatment while minimizing adverse effects and reducing the risk of recurrent infections.

## Supporting information

**S1 File.  S1 Fig.** Schematic of numerical method for diffusion. **S2 Text** Description of model derivation. **S3.1 Fig.** Mesh independent study results where x is the location in the tissue away from the bone plate, and Δx is the mesh size. **S3.1 Table** Mesh quality of the fluid domain in case of bone plate. **S3.2 Fig.** Mesh of fluid domain in case of bone plate. **S4 Fig.** The distribution of scaling factors in the sensitivity analysis.
(DOCX)

## Acknowledgments

The authors would like to express their gratitude to Professor Peter L. Douglas and Professor Eric Croiset for their valuable comments and recommendations that enhanced the quality of our simulation model.

## Author Contributions

**Conceptualization:** Papon Thamvasupong, Kwanchanok Viravaidya-Pasuwat.

**Data curation:** Papon Thamvasupong.

**Funding acquisition:** Kwanchanok Viravaidya-Pasuwat.

**Investigation:** Papon Thamvasupong.

**Methodology:** Papon Thamvasupong.

**Supervision:** Kwanchanok Viravaidya-Pasuwat.

**Validation:** Papon Thamvasupong.

**Writing – original draft:** Papon Thamvasupong.

**Writing – review & editing:** Kwanchanok Viravaidya-Pasuwat.

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
