## [Decision Letter · Decision Letter 0]

2 Jul 2024

PONE-D-24-01184Mathematical Modeling of Vancomycin Release from Poly-L-Lactic Acid-Coated ImplantsPLOS ONE

Dear Dr. Viravaidya-Pasuwat,

Thank you for submitting your manuscript to PLOS ONE. After careful consideration, we feel that it has merit but does not fully meet PLOS ONE’s publication criteria as it currently stands. Therefore, we invite you to submit a revised version of the manuscript that addresses the points raised during the review process.

**ACADEMIC EDITOR:  Please revise based on the feedback below**

We look forward to receiving your revised manuscript.

Kind regards,

Alessio Alexiadis

Academic Editor

PLOS ONE

“This work was supported by King Mongkut’s University of Technology Thonburi under grant the Petchra Pra Jom Klao Ph.D. Research Scholarship (5/2015); the National Research Council of Thailand under FY2017 Thesis Grants for Doctoral Degree Students (NRCT(A)(DPARB) 8/2017, Appendix 7, No. 4 on the grantee list); and Thailand Science Research and Innovation under grant Fundamental Fund-Basic Research Fund (64-3840-FF).”

“This work was supported by King Mongkut’s University of Technology Thonburi under grant the Petchra Pra Jom Klao Ph.D. Research Scholarship (5/2015); the National Research Council of Thailand under FY2017 Thesis Grants for Doctoral Degree Students (NRCT(A)(DPARB) 8/2017, Appendix 7, No. 4 on the grantee list); and Thailand Science Research and Innovation under grant Fundamental Fund-Basic Research Fund (64-3840-FF).”

“This work was supported by King Mongkut’s University of Technology Thonburi under grant the Petchra Pra Jom Klao Ph.D. Research Scholarship (5/2015); the National Research Council of Thailand under FY2017 Thesis Grants for Doctoral Degree Students (NRCT(A)(DPARB) 8/2017, Appendix 7, No. 4 on the grantee list); and Thailand Science Research and Innovation under grant Fundamental Fund-Basic Research Fund (64-3840-FF).”

5. We note that your Data Availability Statement is currently as follows: [All relevant data are within the manuscript and its supporting information files.]

Reviewers' comments:

Reviewer's Responses to Questions

**Comments to the Author**

1. Is the manuscript technically sound, and do the data support the conclusions?

Reviewer #1: Partly

Reviewer #2: Yes

Reviewer #3: No

2. Has the statistical analysis been performed appropriately and rigorously? 

Reviewer #1: Yes

Reviewer #2: N/A

Reviewer #3: Yes

3. Have the authors made all data underlying the findings in their manuscript fully available?

Reviewer #1: Yes

Reviewer #2: Yes

Reviewer #3: Yes

4. Is the manuscript presented in an intelligible fashion and written in standard English?

Reviewer #1: Yes

Reviewer #2: Yes

Reviewer #3: Yes

5. Review Comments to the Author

Reviewer #1: In this MS, Papon Thamvasupong, and Kwanchanok Viravaidya-Pasuwat provide a Mathematical Modeling of Vancomycin Release from PLLA-Coated Implants. Simply, the presentation of the MS is excellent and in general, the MS is a well-written. Clarification and suggestion of a few points, as listed below, would be helpful in order to boost the clarity and impact of this MS.

(i) Single-dose kinetics of intravenous vancomycin has a bi-phasic elimination half-life, with its initial half-life being relatively quick and a terminal half-life of 4.7 to 11.2 hours in healthy adults with normal renal function(Krogstad DJ, 1980 PMID: 7381031). The elimination half-life is significantly prolonged in patients with renal dysfunction. Close Therapeutic drug monitoring (TDM) is necessary for these patients. Another article reported that the half-life of the initial phase is approximately 7 minutes, that of the second phase is approximately 0.5 to 1 hour, while the terminal elimination half-life ranges from 3 to 9 hours in subjects with normal renal function (Matzke GR, 1986 PMID: 3530582). Moreover, excessively high serum concentrations of vancomycin is associated with ototoxicity (Moellering RC Jr. 1984 PMID: 6394577). Vancomycin disposition is also dependent on age (estimated t1/2 obtained from the terminal phase is longer in the elderly than in the normal healthy men (12.1 and 7.2 hr) at an intravenous infusion of 6 mg/kg (Cutler NR, 1984 PMID: 6499360). Finally suppression of bone marrow progenitor cell growth by vancomycin following autologous stem cell transplantation. Meehan KR, 1997 PMID: 9169648. Thus, single-dose of vancomycin in implants should be less than 15 mg/kg in preoperative antimicrobial prophylaxis or surgical prophylaxis (Off-label)

In page 17 line –“Nonetheless, the vancomycin concentration remained below the toxicity threshold of 1,000 μg/mL “ remains obscure in this sense and need to clarify the half-life and dose dependent manner

(ii) The rate and extent of penetration of an antibacterial into bone determine its therapeutic success. However, bone tissue consists of an organic fraction consisting of mainly of collagen fibrils, glycoproteins, proteoglycans and extracellular fluid and an inorganic fraction consisting mainly of calcium phosphate as hydroxyapatite crystals as well as bone cells. The affinity of the antimicrobial for hydroxyapatite as the main constituent of the inorganic matter needs to be considered for some antibiotics including the quinolones ofloxacin, ciprofloxacin and pefloxacin. Imipenem is also found to bind to hydroxyapatite and to human bone to a minor extent. Fosfomycin has been shown to bind to hydroxyapatite (Landersdorfer CB, 2009 PMID: 19271782; Thabit AK, 2019 PMID: 30772469).

Is it possible to find out the interaction between vancomycin and hydroxyapatite crystal composition?

Reviewer #2: The manuscript by Thammasupong and Viravaidya-Pasuwat presents the development of a mathematical model to predict the release profile and antibacterial efficacy of a vancomycin delivery system designed for bone plates. The manuscript demonstrates commendable organization and provides valuable insights into biomedical materials. However, some minor considerations should be addressed before publication:

1. The main quantitative results should be reported in the abstract section.

2. A brief methodology on the use of the finite element software ANSYS-180 CFX 2021R package should be included.

3. The authors indicated that they employed data from their previous research

(https://doi.org/10.3390/polym14173493) to construct the proposed mathematical model. However, they did not simulate the release profiles of vancomycin reported in that work. Given that the simulation of the release profiles is a crucial validation step for the mathematical model, it is essential for the authors to complete this task. The inclusion of these simulations would not only strengthen the reliability of the model but also enhance the overall contribution of the manuscript to the field. It would provide a more comprehensive understanding of the model's predictive capabilities and its applicability in real-world scenarios. Therefore, the authors should conduct and report the simulations of the vancomycin release profiles to ensure the robustness and applicability of their model.

Reviewer #3: Revision about manuscript ID PONE-D-24-01184

The manuscript provides a comprehensive computational modeling approach to study the local delivery of vancomycin from a bone plate. The validation of the model using experimental data is supposed to increase the confidence in the simulation results. The sensitivity analysis on the key parameters, such as vancomycin half-life and diffusion coefficients, provides valuable insights into the factors that influence the drug distribution. English is very well written. However, the manuscript could be improved by providing more details on the model development and the assumptions made, as well as a more thorough discussion of the limitations and potential applications of the approach.

The abstract gives the idea that this manuscript performed both mathematical and animal model and compared the results. You should perform a clear text depicting that the comparison is from previous literature already published.

The work was strongly compared to the animal model previous published by Loc-Carrillo et al (2016), reference 14. I have read this literature, and I found some inconsistences in the comparison made when presented in the figure 6 of your manuscript. Additionally, it´s not clear in this figure from which reference you obtained the comparison of the animal study. However, in lines 92-95 and 283-84, you give a support text that bring indicative discussion that Figure 6 was performed using this literature for comparison. The authors should be clear in the description of figure 6 legend, to describe from which reference obtained this comparison. Furthermore, after reading the reference 14, I observed that their release profile in vivo presents a heterogenic delivery while in your Figure 6, you presented a continuous homogenous delivery, strongly correlated to your mathematical model. Other important point is that the units obtained in the delivery profile of reference 14 is different (ug/g) from the units you used (ug/mL), how you converted units? Did you consider g and mL as the same thing? Did the research group of reference 14 gave you the converted data? With so, I suggest their citation in acknowledgments. You should explain with more details how obtained the data of your Figure 6 from Fig 1A of Loc-Carrilo (2016). In the topic Cross-Validation with Published in vivo Data you discuss about this complexity and data refinement. Thus, I strongly recommend that details about how you processed the data comparison to a very similar mathematical model relating to a literature that presents a different delivery pattern should more well described since this comparison is the most important data of your work.

Despite other minor details on final conclusions, which depicts about “patient safety” in general, the target of this model is to predict the drug effectiveness while when in a delivery PLLA based vancomycin model, thus I recommend depicting more direct outcome such as effective treatment minimizing adverse effects and to avoid ineffective treatment leading to recurrent infection.

After the corrections about the discussion related to figure 6 and abstract rewritten, I recommend to manuscript to be published.

6. PLOS authors have the option to publish the peer review history of their article (what does this mean?). If published, this will include your full peer review and any attached files.

Reviewer #1: **Yes: **Dr. Abdul Alim Al-Bari

Reviewer #2: No

Reviewer #3: No

---

## [Author Response · Author response to Decision Letter 0]

2 Sep 2024

Reviewer #1: In this MS, Papon Thamvasupong, and Kwanchanok Viravaidya-Pasuwat provide a Mathematical Modeling of Vancomycin Release from PLLA-Coated Implants. Simply, the presentation of the MS is excellent and in general, the MS is a well-written. Clarification and suggestion of a few points, as listed below, would be helpful in order to boost the clarity and impact of this MS.

(i) Single-dose kinetics of intravenous vancomycin has a bi-phasic elimination half-life, with its initial half-life being relatively quick and a terminal half-life of 4.7 to 11.2 hours in healthy adults with normal renal function (Krogstad DJ, 1980 PMID: 7381031). The elimination half-life is significantly prolonged in patients with renal dysfunction. Close Therapeutic drug monitoring (TDM) is necessary for these patients. Another article reported that the half-life of the initial phase is approximately 7 minutes, that of the second phase is approximately 0.5 to 1 hour, while the terminal elimination half-life ranges from 3 to 9 hours in subjects with normal renal function (Matzke GR, 1986 PMID: 3530582). Moreover, excessively high serum concentrations of vancomycin is associated with ototoxicity (Moellering RC Jr. 1984 PMID: 6394577). Vancomycin disposition is also dependent on age (estimated t1/2 obtained from the terminal phase is longer in the elderly than in the normal healthy men (12.1 and 7.2 hr) at an intravenous infusion of 6 mg/kg (Cutler NR, 1984 PMID: 6499360). Finally suppression of bone marrow progenitor cell growth by vancomycin following autologous stem cell transplantation. Meehan KR, 1997 PMID: 9169648. Thus, single-dose of vancomycin in implants should be less than 15 mg/kg in preoperative antimicrobial prophylaxis or surgical prophylaxis (Off-label)

In page 17 line –“Nonetheless, the vancomycin concentration remained below the toxicity threshold of 1,000 μg/mL “ remains obscure in this sense and need to clarify the half-life and dose dependent manner

Response: Thank you for your thorough review and for emphasizing the critical aspects of vancomycin pharmacokinetics, particularly its multi-phasic elimination half-life and the importance of dose-dependent toxicity. We have incorporated your suggestions and citations in the revised manuscript, specifically in the Results and Discussion section (page 19, lines 392 – 405). In addition, we have expanded the analysis of vancomycin toxicity, addressing the concerns related to dose dependency on page 18, lines 360 – 376. We believe these revisions enhance the clarity and depth of the manuscript.

(ii) The rate and extent of penetration of an antibacterial into bone determine its therapeutic success. However, bone tissue consists of an organic fraction consisting of mainly of collagen fibrils, glycoproteins, proteoglycans and extracellular fluid and an inorganic fraction consisting mainly of calcium phosphate as hydroxyapatite crystals as well as bone cells. The affinity of the antimicrobial for hydroxyapatite as the main constituent of the inorganic matter needs to be considered for some antibiotics including the quinolones ofloxacin, ciprofloxacin and pefloxacin. Imipenem is also found to bind to hydroxyapatite and to human bone to a minor extent. Fosfomycin has been shown to bind to hydroxyapatite (Landersdorfer CB, 2009 PMID: 19271782; Thabit AK, 2019 PMID: 30772469).

Is it possible to find out the interaction between vancomycin and hydroxyapatite crystal composition?

Response: Thank you for your insightful comment regarding the interaction between vancomycin and hydroxyapatite. We have identified a study that demonstrates vancomycin’s adsorption onto hydroxyapatite, followed by its gradual release into the surrounding fluid. We believe that the binding of vancomycin to hydroxyapatite could influence the concentration profile predicted by our simulation. However, based on the reference the reviewer suggested (Thabit AK, 2019 PMID: 30772469), vancomycin still exhibits a favorable penetration profile into bone, suggesting that this interaction may not significantly alter the predicted concentration profile as predicted. Nevertheless, we agree that integrating this interaction into our model could enhance its accuracy. We have included this discussion and the relevant references in the revised manuscript on page 21, lines 435 – 448. 

Reviewer #2: The manuscript by Thammasupong and Viravaidya-Pasuwat presents the development of a mathematical model to predict the release profile and antibacterial efficacy of a vancomycin delivery system designed for bone plates. The manuscript demonstrates commendable organization and provides valuable insights into biomedical materials. However, some minor considerations should be addressed before publication:

1. The main quantitative results should be reported in the abstract section.

Response: Thank you for your recommendation. We have updated the abstract to include key quantitative results, including the maximum concentration of vancomycin and the sensitivity indices from the sensitivity analysis on page 2, lines 24 – 33 of our revised manuscript. 

2. A brief methodology on the use of the finite element software ANSYS-180 CFX 2021R package should be included.

Response: We have added a brief description of the methodology on the use of ANSYS-CFX 2021R on page 10, lines 180 – 192 of the revised manuscript. 

3. The authors indicated that they employed data from their previous research

(https://doi.org/10.3390/polym14173493) to construct the proposed mathematical model. However, they did not simulate the release profiles of vancomycin reported in that work. Given that the simulation of the release profiles is a crucial validation step for the mathematical model, it is essential for the authors to complete this task. The inclusion of these simulations would not only strengthen the reliability of the model but also enhance the overall contribution of the manuscript to the field. It would provide a more comprehensive understanding of the model's predictive capabilities and its applicability in real-world scenarios. Therefore, the authors should conduct and report the simulations of the vancomycin release profiles to ensure the robustness and applicability of their model. 

Response: Thank you for your recommendation. We agree that validating our model against our previous in vitro vancomycin release study would strengthen our manuscript. We have conducted simulation to replicate the vancomycin release from the double-layered PLLA coated substrate. The model assumptions are provided in the Materials and Methods section on page 10 – 11, lines 201 – 208 and the validation result is presented in Fig. 4a with the corresponding discussion on page 15, lines 289 – 294 of the revised manuscript.

Reviewer #3: Revision about manuscript ID PONE-D-24-01184

The manuscript provides a comprehensive computational modeling approach to study the local delivery of vancomycin from a bone plate. The validation of the model using experimental data is supposed to increase the confidence in the simulation results. The sensitivity analysis on the key parameters, such as vancomycin half-life and diffusion coefficients, provides valuable insights into the factors that influence the drug distribution. English is very well written. 

1. However, the manuscript could be improved by providing more details on the model development and the assumptions made, as well as a more thorough discussion of the limitations and potential applications of the approach.

Response: Thank you for your insightful suggestion. We have added more detailed information on the use of ANSYS-CFX 2021R and the assumptions made during model development on page 10, lines 180 – 192 of the revised manuscript. We also added more discussion on the potential effects of vancomycin half-life (page 19, lines 392 – 405) and the interaction between vancomycin and hydroxyapatite in bone (page 21, lines 435 – 448) on the model prediction. The limitation section has been updated accordingly (page 23, lines 489 – 493). 

As noted in previous studies, vancomycin pharmacokinetics should be described by a multi-phase profile, with half-lives varying significantly depending on factors such as age and medical history. These variations can substantially influence our model, which currently uses a simplified half-life function. Moreover, vancomycin binding to hydroxyapatite may alter the predicted concentration profile, leading to potential under- or over-estimations. Despite these limitations, our model still offers valuable insights into vancomycin release from the bone plate to surrounding tissues. For improved accuracy, more detailed vancomycin pharmacokinetics should be integrated into our model.

2. The abstract gives the idea that this manuscript performed both mathematical and animal model and compared the results. You should perform a clear text depicting that the comparison is from previous literature already published.

Response: We apologize for the confusion. To clarify, we have revised the abstract to explicitly state that the comparison is made with data from previously published literature. This clarification can be found on page 2, line 25 of our revised manuscript.

3. The work was strongly compared to the animal model previous published by Loc-Carrillo et al (2016), reference 14. I have read this literature, and I found some inconsistences in the comparison made when presented in the figure 6 of your manuscript. Additionally, it´s not clear in this figure from which reference you obtained the comparison of the animal study. However, in lines 92-95 and 283-84, you give a support text that bring indicative discussion that Figure 6 was performed using this literature for comparison. The authors should be clear in the description of figure 6 legend, to describe from which reference obtained this comparison. 

Response: Once again, we apologize for the confusion. We have revised the legend of Fig. 5 to clearly state that the in vivo experimental data used for comparison with our simulation model are derived from previously published literature (page 33, lines 724 – 726).

4. Furthermore, after reading the reference 14, I observed that their release profile in vivo presents a heterogenic delivery while in your Figure 6, you presented a continuous homogenous delivery, strongly correlated to your mathematical model. Other important point is that the units obtained in the delivery profile of reference 14 is different (ug/g) from the units you used (ug/mL), how you converted units? Did you consider g and mL as the same thing? Did the research group of reference 14 gave you the converted data? With so, I suggest their citation in acknowledgments. You should explain with more details how obtained the data of your Figure 6 from Fig 1A of Loc-Carrilo (2016). In the topic Cross-Validation with Published in vivo Data you discuss about this complexity and data refinement. Thus, I strongly recommend that details about how you processed the data comparison to a very similar mathematical model relating to a literature that presents a different delivery pattern should more well described since this comparison is the most important data of your work.

Response: Thank you for pointing out the areas that needed clarification. We acknowledge that our manuscript did not fully explain how we obtained and processed the in vivo data from Loc-Carrilo et al. (2016) for comparison of our model. The raw data of Figure 1A of Loc-Carrilo, et al. are provided in the Supporting Information S1 of their manuscript, as shown in the table below. 

 We have now cited this source explicitly in the Materials and Methods section on page 11, lines 224 – 226 of our revised manuscript. In addition, we have modified Fig. 5 to accurately represent the experimental data from Loc-Carrilo, et al. Instead of using a dotted line, which might suggest a continuous release profile, we now use bubble points to indicate the specific data points at various time intervals to reflect the heterogeneity in the profile observed in the original study. 

5. Despite other minor details on final conclusions, which depicts about “patient safety” in general, the target of this model is to predict the drug effectiveness while when in a delivery PLLA based vancomycin model, thus I recommend depicting more direct outcome such as effective treatment minimizing adverse effects and to avoid ineffective treatment leading to recurrent infection.

After the corrections about the discussion related to figure 6 and abstract rewritten, I recommend to manuscript to be published.

Response: Thank you for your suggestion. We have replaced the phrase “patient safety” with a more specific statement that focuses on effective treatment, minimizing adverse effects, and avoiding recurrent infections, as shown on page 24, lines 512 – 514 of our revised manuscript.

---

## [Decision Letter · Decision Letter 1]

20 Sep 2024

Mathematical Modeling of Vancomycin Release from Poly-L-Lactic Acid-Coated Implants

PONE-D-24-01184R1

Dear Dr. Viravaidya-Pasuwat,

We’re pleased to inform you that your manuscript has been judged scientifically suitable for publication and will be formally accepted for publication once it meets all outstanding technical requirements.

Kind regards,

Bashar Ibrahim

Academic Editor

PLOS ONE

Reviewers' comments:

Reviewer's Responses to Questions

**Comments to the Author**

1. If the authors have adequately addressed your comments raised in a previous round of review and you feel that this manuscript is now acceptable for publication, you may indicate that here to bypass the “Comments to the Author” section, enter your conflict of interest statement in the “Confidential to Editor” section, and submit your "Accept" recommendation.

Reviewer #1: All comments have been addressed

Reviewer #2: All comments have been addressed

2. Is the manuscript technically sound, and do the data support the conclusions?

Reviewer #1: Yes

Reviewer #2: Yes

3. Has the statistical analysis been performed appropriately and rigorously? 

Reviewer #1: Yes

Reviewer #2: Yes

4. Have the authors made all data underlying the findings in their manuscript fully available?

Reviewer #1: Yes

Reviewer #2: Yes

5. Is the manuscript presented in an intelligible fashion and written in standard English?

Reviewer #1: Yes

Reviewer #2: Yes

6. Review Comments to the Author

Reviewer #1: I am satisfied to your response based on comments.

"Single-dose of vancomycin

in implants should be less than 15 mg/kg in preoperative antimicrobial prophylaxis or surgical prophylaxis (Off-label)" is interesting

Reviewer #2: In my opinion, the latest revision of the manuscript has significantly improved, addressing the previous concerns effectively. As it now meets the necessary standards of clarity, rigor, and completeness, I believe it is well-suited for publication

7. PLOS authors have the option to publish the peer review history of their article (what does this mean?). If published, this will include your full peer review and any attached files.

Reviewer #1: **Yes: **Dr. Abdul Alim Al-Bari

Reviewer #2: No

---

## [Editor Report · Acceptance letter]

27 Sep 2024

PONE-D-24-01184R1 

PLOS ONE

Dear Dr. Viravaidya-Pasuwat, 

I'm pleased to inform you that your manuscript has been deemed suitable for publication in PLOS ONE. Congratulations! Your manuscript is now being handed over to our production team.

Kind regards, 

on behalf of

Prof. Dr. Bashar Ibrahim 

Academic Editor

PLOS ONE
